# Development of Polymer-Encapsulated, Amine-Functionalized Zinc Ferrite Nanoparticles as MRI Contrast Agents

**DOI:** 10.3390/ijms242216203

**Published:** 2023-11-11

**Authors:** Ágnes M. Ilosvai, László Forgách, Noémi Kovács, Fatemeh Heydari, Krisztián Szigeti, Domokos Máthé, Ferenc Kristály, Lajos Daróczi, Zoltán Kaleta, Béla Viskolcz, Miklós Nagy, László Vanyorek

**Affiliations:** 1Institute of Chemistry, University of Miskolc, 3515 Miskolc, Hungary; maria.agnes.ilosvai@uni-miskolc.hu (Á.M.I.); bela.viskolcz@uni-miskolc.hu (B.V.); miklos.nagy@uni-miskolc.hu (M.N.); 2Higher Education and Industrial Cooperation Centre, University of Miskolc, 3515 Miskolc, Hungary; 3Department of Biophysics and Radiation Biology, Semmelweis University, 1094 Budapest, Hungary; noemi.kovacs@hcemm.eu (N.K.); fatemeh.heydari@stud.semmelweis.hu (F.H.); szigeti.krisztian@med.semmelweis-univ.hu (K.S.); domokos.mathe@hcemm.eu (D.M.); 4In Vivo Imaging Advanced Core Facility, Hungarian Center of Excellence for Molecular Medicine (HCEMM), 1094 Budapest, Hungary; 5Institute of Mineralogy and Geology, University of Miskolc, 3515 Miskolc, Hungary; askkf@uni-miskolc.hu; 6Department of Solid State Physics, University of Debrecen, P.O. Box 2, 4010 Debrecen, Hungary; lajos.daroczi@science.uniceb.hu; 7Pro-Research Laboratory, Progressio Engineering Bureau Ltd., 8000 Szekesfehervar, Hungary; kaleta.zoltan@semmelweis.hu; 8Institute of Organic Chemistry, Semmelweis University, 1092 Budapest, Hungary

**Keywords:** fast microwave synthesis, MRI, magnetic nanoparticles, ferrite, zinc

## Abstract

The need for stable and well-defined magnetic nanoparticles is constantly increasing in biomedical applications; however, their preparation remains challenging. We used two different solvothermal methods (12 h reflux and a 4 min microwave, MW) to synthesize amine-functionalized zinc ferrite (ZnFe_2_O_4_-NH_2_) superparamagnetic nanoparticles. The morphological features of the two ferrite samples were the same, but the average particle size was slightly larger in the case of MW activation: 47 ± 14 nm (Refl.) vs. 63 ± 20 nm (MW). Phase identification measurements confirmed the exclusive presence of zinc ferrite with virtually the same magnetic properties. The Refl. samples had a zeta potential of −23.8 ± 4.4 mV, in contrast to the +7.6 ± 6.8 mV measured for the MW sample. To overcome stability problems in the colloidal phase, the ferrite nanoparticles were embedded in polyvinylpyrrolidone and could be easily redispersed in water. Two PVP-coated zinc ferrite samples were administered (1 mg/mL ZnFe_2_O_4_) in X BalbC mice and were compared as contrast agents in magnetic resonance imaging (MRI). After determining the r1/r2 ratio, the samples were compared to other commercially available contrast agents. Consistent with other SPION nanoparticles, our sample exhibits a concentrated presence in the hepatic region of the animals, with comparable biodistribution and pharmacokinetics suspected. Moreover, a small dose of 1.3 mg/body weight kg was found to be sufficient for effective imaging. It should also be noted that no toxic side effects were observed, making ZnFe_2_O_4_-NH_2_ advantageous for pharmaceutical formulations.

## 1. Introduction

The general use of ferrites includes applications in electronics, telecommunications, power generation, magnetic storage media, electromagnetic interference (EMI) suppression, and microwave absorbers [1,2,3,4]. Their excellent magnetic properties, high electrical resistivity, and relatively low cost make them essential components in a wide range of industrial and technological products. In recent years, there has been a growing interest in exploring ferrites for biomedical applications, particularly as MRI contrast agents and in targeted drug delivery systems, owing to their biocompatibility and potential for the controlled release of therapeutic agents, hyperthermia treatment, and isolation of macromolecules, for example, deoxyribonucleic acid (DNA) [5,6,7,8,9,10,11,12,13].

In contemporary clinical practice, gadolinium-chelates are the predominant choice for contrast enhancement in magnetic resonance imaging (MRI) examinations [14,15]. The use of Gd-based contrast agents, however, raises toxicity issues due to the unexpected release of free Gd ions in the body [16,17]. Magnetic nanoparticles, especially superparamagnetic ones, including magnetite, maghemite, and the various ferrite nanoparticles, are promising alternatives to MRI contrast agents due to their improved specificity and biocompatibility [18,19]. Particular care must be taken to ensure that the contrast agents developed are nontoxic. Several previously authorized formulations were found to cause adverse side effects (local pain, hypotension, hypersensitivity, anaphylactic shock, vasodilatation, and paresthesia) [20]. These nanomaterial-based contrast agents were withdrawn from use in 2008/2009, including ferumoxide (Feridex^®^, Endorem^®^) and ferucarbotran (Ciavist™, Resovist^®^) used for liver imaging [20,21,22]. Iron oxide based nanoparticle (IONP) contrast agents should be used in the lowest possible concentration, due to iron overload resulting in the increased production of reactive oxygen species (ROS) via Fenton and Haber–Weiss reactions, which can cause intracellular damage [23,24,25,26]. Nevertheless, cells can effectively regulate iron metabolism through various antioxidant defense mechanisms, such as the antioxidant glutathione and the enzymes glutathione peroxidase and glutathione reductase, or by storing excess iron in ferritin [26,27]. However, in the case of overload, these various antioxidant mechanisms will not be able to effectively reactive oxygen species (e.g., OH radicals), resulting in intracellular damage (e.g., lipid peroxidation, protein oxidation, DNA damage, and then cell death) [28,29]. Therefore, it is necessary to develop magnetic nanoparticles that can be used effectively at the lowest possible concentration, with reduced toxicity. The toxicity of magnetic nanoparticles can also be reduced by adding biocompatible layers on the surface, as Ganguly et al. demonstrated: a layer of human serum albumin (HSA) was deposited on the surface of maghemite NPs functionalized with iohexol. Furthermore, the presence of HSA and iohexol enabled the development of a multifunctional (trimodal) contrast agent that penetrates in the cells and results in an excellent signal in several types of imaging diagnostics methods, i.e., fluorescence imaging (FI), computed tomography (CT), and magnetic resonance imaging (MRI) [30,31].

Iron oxide nanoparticles have advantages such as an integration into iron metabolism and an easy surface functionalization with target molecules, enabling molecular imaging for disease identification [32,33]. Targeted magnetic nanoparticles exhibit the ability to accumulate within a tumor 24 h after intravenous injection, demonstrating their efficiency as contrast agents for generating T2-weighted MR images. They are extensively used in the diagnosis of diverse cancer types, such as breast cancer, pancreatic cancer, and glioblastoma multiforme (GBM) [34,35,36,37,38]. For the abovementioned application, ferrite nanoparticles are well suited as alternatives to magnetite and maghemite. These magnetic nanomaterials are spinel-type metal oxides with the chemical composition of MFe_2_O_4_ (M = Mn^2+^, Ni^2+^, Co^2+^, Mg^2+^, Zn^2+^ and Cd^2+^) [39,40,41]. In this sense, these materials consist of a combination of iron oxide (Fe_2_O_3_) and other metal oxides, such as zinc, manganese, nickel, or cobalt. Not only can ferrite nanoparticles differ in composition but also in preparation techniques and synthesis time.

Various methods for the synthesis of magnetic nanoparticles are available in the literature [42,43,44]. The special properties of ferrite nanoparticles allow for direct functionalization with amino groups [45] or different coatings [46] on their surface. Chattharika Aoopngan et al. used a 12 h coprecipitation method to synthesize magnesium ferrite nanoparticles functionalized with amine [45]. A widely used solvothermal method to produce ferrite nanoparticles occurs in a mixture of ethylene glycol (EG),ethanol and EG-water in the presence of surfactants and sodium acetate. These synthesis methods also yield nanoparticles with similar morphologies [47,48,49].

Another synthesis method is the coprecipitation of Mn-Zn nanoparticles doped with a rare earth metal (Sm, samarium) conducted by N. Yadav et al. [50]. Nevertheless, ferrites can be produced not only by coprecipitation but also via sonochemical [51], microemulsion [52], or hydrothermal synthesis [53].

Recently, researchers have become increasingly interested in the microwave-assisted (MW) synthesis of nanomaterials due to its favorable attributes, such as environmentally friendly reactions, shorter reaction times, efficient volumetric heating, energy conservation, and a high reaction rate. Microwave energy transforms into heat within the material, resulting in reduced energy consumption, shorter processing time, and precise volumetric heating with a specific temperature distribution [54].

An efficient and rapid method for the synthesis of ferrite nanoparticles is microwave-induced combustion, which was used by Yalçıner et al. to produce nickel ferrite [55]. The process involved dissolving metal precursors and glycine in water and heating the reaction mixture in a microwave oven. Initially, the solution boils and evaporates, followed by the decomposition of the reactants (glycine and nitrate salts). During the burning of the mixture, the glycine serves as fuel, being oxidized by nitrate. Shukla et al. fabricated titanium-ferrite nanoparticles using a 15 min MW synthesis, while Sertkol et al. fabricated Zn-doped nickel-ferrite nanoparticles that were subjected to microwave exposure for only 5 min [56,57]. Zinc ferrite nanoparticles are a great example for materials as such. In this study, attention was directed towards the synthesis of amine-functionalized zinc ferrite nanoparticles. Two distinct methods were employed: a conventional 12 h coprecipitation process with monoethanolamine in ethylene glycol and a novel microwave-assisted synthesis, reducing the reaction time significantly to four minutes. While the conventional 12 h synthesis method is known for producing high-quality zinc ferrite nanoparticles suitable for biomedical applications, the novel microwave-assisted synthesis offers a rapid and energy-efficient alternative. However, concerns have been raised regarding the potential distortion in the magnetic properties of nanoparticles synthesized using the microwave-assisted method. Given the critical importance of maintaining optimal magnetic properties in nanoparticles intended for biomedical use, this study seeks to investigate and compare the effects of both synthesis methods on the magnetic characteristics of zinc ferrite nanoparticles. The evaluation of the magnetic properties is crucial to determine the feasibility of employing the microwave-assisted synthesis in the biomedical field as an MRI contrast agent. Through this research, we aim to strike a balance between the efficiency-driven requirements of industrial applications, where cost reduction is significant, and the stringent standards necessary for biomedical applications, where magnetic properties are of utmost importance. The findings of this study will shed light on the potential impact of the microwave-assisted synthesis on the nanoparticles’ suitability for biomedical use, ultimately contributing to the understanding and optimization of zinc ferrite nanoparticles for diverse applications in both industrial and biomedical fields.

## 2. Results

### 2.1. Preparation and Characterization of the Amine-Functionalized ZnFe_2_O_4_ Samples

Amine-functionalized ZnFe_2_O_4_ spinels were synthesized by a modified coprecipitation method, employing two different synthesis routes (Figure 1). The ZnFe_2_O_4_-NH_2_ Refl. Sample was prepared by heating under reflux for 12 h (referred to as synthesis method “A” or “Refl.”). The reflux in ethylene glycol (EG) is lengthy and energy-consuming; therefore, microwave synthesis was utilized to prepare amine-functionalized zinc ferrite, resulting in a notable reduction of the synthesis time to only 4 min (referred to as synthesis method “B” or “MW”). In the following, we will refer to the samples prepared by method A and B as ZnFe_2_O_4_-NH_2_ Refl. And ZnFe_2_O_4_-NH_2_ MW, respectively.

The formation of zinc ferrite nanoparticles is assisted by the transformation of MEA because the presence of zinc(II) ions reduces the thermal decomposition temperature of MEA [58], and ammonia is released. Furthermore, Rochell has shown that the presence of Fe(III) ions promotes the oxidative degradation of MEA; during this process, aminium radicals are formed which are deprotonated to imine radicals which are further degraded to ammonia and aldehyde (hydroxyacetaldehyde) via imine [59]. The alkaline conditions lead to the formation of Fe(OH)_3_ and Zn(OH)_2_, which can be transformed into ZnFe_2_O_4_ through dehydration.
$${NH}_{2}{CH}_{2}{CH}_{2}OH\overset{{Fe}^{3+}{and-H}^{+}}{\to }HN{=CH-CH}_{2}-OH+H_{2}O\overset{{Fe}^{3+}}{\to }HO{CH}_{2}CHO+{NH}_{3}$$
$${NH}_{3}+H_{2}O\to {NH}_{4}^{+}+{OH}^{-}$$
$${Fe}^{3+}+3{OH}^{-}\to {Fe(OH)}_{3}$$
$${Zn}^{2+}+{2OH}^{-}\to {Zn(OH)}_{2}$$
$${Zn(OH)}_{2}+{2Fe(OH)}_{3}\to {ZnFe}_{2}O_{4}+{4H}_{2}O$$

The morphology and particle size of the amine-functionalized zinc ferrite crystallites were investigated by high-resolution transmission electron microscopy (HRTEM) (Figure 2). The HRTEM pictures clearly show the spherical shape of the functionalized ferrite nanoparticles, which are composed of smaller individual nanoparticles measuring 5–15 nm in size (Figure 2D–F). The average crystallite size of the aforementioned individual ferrite crystallites was also calculated based on XRD measurements. The sizes of the individual nanoparticles, as observed in the HRTEM pictures, are consistent with the XRD measurements. The average size of the nanoparticles (based on the XRD results) in the case of the ZnFe_2_O_4_-NH_2_ Refl. Sample is 9 ± 2 nm. In the case of the ZnFe_2_O_4_-NH_2_ MW nanospheres (building blocks), the average size of the individual nanoparticles was 12 ± 2 nm.

In contrast, the average particle sizes of the aggregated ferrite spheres, as determined from HRTEM pictures, in the case of the ZnFe_2_O_4_-NH_2_ Refl. And ZnFe_2_O_4_-NH_2_ MW samples are 47 ± 14 nm and 63 ± 20 nm, respectively (Table 1 and Figure 2C). In the case of microwave-assisted synthesized ferrite, particles with larger sizes are observed, which is also reflected in the increased minimum, maximum, P90, and P95 values.

Based on the results of selected area electron diffraction (SAED), the nanoparticles in the HRTEM pictures were identified as ZnFe_2_O_4_ spinels for both -NH_2_-functionalized samples (Figure 3A, D). The measured d-spacing values were correlated with the d-values in X-ray databases, specifically the powder diffraction file of the franklinite (PDF 22-1012) spinel structure (Figure 3B,E). In the case of ZnFe_2_O_4_ samples produced by microwave synthesis and reflux treatment, the same reflections were identified on the diffractograms, which were consistent with the SAED results (Figure 3C). The reflection peaks were identified at 29.9° (220), 35.3° (311), 36.7° (222), 42.9° (400), 53.2° (422), 56.7° (511), and 62.7° (440) 2 Theta degrees (PDF 22-1012). Each of these peaks corresponds to a single metal oxide phase, ZnFe_2_O_4_, indicating the presence of pure franklinite (Figure 3C). This conclusion holds true for both samples produced using different synthesis methods.

The surface functional groups of the amine-functionalized ferrite nanoparticles were identified using FTIR. In the FTIR spectra of the amine-functionalized ZnFe_2_O_4_ samples, two characteristic peaks were observed at wavenumbers of 430 cm^−1^ and 580 cm^−1^ (Figure 4A). These peaks were assigned to the intrinsic stretching vibration modes of the metal–oxygen bonds at the octahedral and tetrahedral sites [60,61]. Additionally, a peak at 890 cm^−1^ was identified as the rocking vibration mode of the -CH_2_ groups, originating from the adsorbed ethylene glycol and monoethanolamine molecules on the surface of the ferrite nanoparticles [60].

In the wavenumber range between 1000 and 1100 cm^−1^, absorption bands were found corresponding to the stretching vibrations of νC-O and νC-N, which belong to the hydroxyl, carboxyl, and amine functional groups. Other visible bands at 1381 cm^−1^ and 1567 cm^−1^ were characteristic of the bending vibration mode of hydroxyl groups (from the adsorbed EG molecules) and the νC=C vibration of adsorbed organic compounds. The band at 1602 cm^−1^ indicated the presence of amine groups due to the adsorbed MEA molecules. Moreover, the band corresponding to the O-H bending vibration of adsorbed water was observed at 1628 cm^−1^ [62,63]. The symmetric and asymmetric stretching vibration of the C-H bonds resulted in absorption at 2859 cm^−1^ and 2929 cm^−1^, which can be explained by the adsorbed organic molecules (EG and MEA) on the surface, indicating the successful surface grafting of ZnFe_2_O_4_. The stretching vibration band of the N–H bonds is convoluted with the vibration band of -OH groups in a broad peak in the 3000–3750 cm^−1^ region. In the case of the zeta potentials, a significant difference was measured because the electrokinetic potential of the ZnFe_2_O_4_-NH_2_ Refl. sample was −23.8 ± 4.4 mV, which is more negative than that of the MW-assisted sample, where +7.6 ± 6.8 mV was obtained (Figure 4B). A fundamental requirement for MRI contrast agents is the sustained stability of the colloid system and the prevention of aggregation. Long-term stability can be ensured by embedding the magnetic nanoparticles in a water-soluble polymer that, after drying, easily dissolves in an aqueous medium and sterically stabilizes the nanoparticles. Poly(vinylpyrrolidone) (PVP) seems to be an ideal choice due to its high-water solubility and biocompatibility.

The magnetization curve of the amine-functionalized ZnFe_2_O_4_ samples was measured at 298 K using a vibrating-sample magnetometer (VSM), as shown in Figure 5A. The experimental curves were fitted by the Langevin function (Equations (1) and (2)), and from the fitting parameters, the saturation magnetization as well as the average magnetic moment of the nanoparticles were evaluated.
(1)$$M\left(H\right)=M_{s}\left(\coth\left(sH\right)-\frac{1}{sH}\right)=M_{s}\left(\frac{e^{sH}+e^{-sH}}{e^{sH}-e^{-sH}}-\frac{1}{sH}\right)$$
(2)$$s=\frac{m\mu_{0}}{kT}$$

Fitting was performed by using a two-parameter nonlinear fit in Origin 2018. The hysteresis-free nature of the experimental curves and the good agreement with the paramagnetic Langevin function indicate that at room temperature the thermal fluctuations significantly overcome the crystal anisotropy energy barrier in the case of these nanoparticles. Moreover, the average crystallite sizes were found to be 9 ± 2 nm and 12 ± 2 nm, as measured by XRD and HRTEM. These very small crystallites are arranged in larger spherical aggregates, visible in the HRTEM images (Figure 2D–F). Superparamagnetism occurs in nanoparticles composed of a single magnetic domain. This is possible when their diameter is below 3–50 nm, depending on the materials [64]; due to the small crystallite sizes of our ZnFe_2_O_4_ particles, they are in the range that allows superparamagnetic behavior.

The magnetic saturation (Ms) values were 16.99 emu/g and 18.35 emu/g in the case of ZnFe_2_O_4_-NH_2_ MW and ZnFe_2_O_4_-NH_2_ Refl., respectively. The magnetic moment (in Bohr magneton units, μB) of the nanoparticles was also calculated: 1064 μB (ZnFe_2_O_4_-NH_2_ MW) and 876 μB (ZnFe_2_O_4_-NH_2_ Refl.) were found. The magnetization curves of the ferrite samples do not exhibit a hysteresis loop. Moreover, the values of the remnant magnetization (Mr) and coercivity (Hc) were approximately zero. Based on these results, it can be stated that the synthesized particles exhibit superparamagnetic behavior at room temperature.

The PVP-encapsulated ferrite nanoparticles easily disperse in an aqueous phase and form a stable colloid, as depicted in Figure 5B. The colloidal system does not show aggregation even after two hours. The PVP-stabilized magnetic nanoparticles can be separated using a neodymium magnet, but once the magnetic field is off again, they redisperse and maintain a stable colloid system.

There are several solvothermal methods in the literature that use EG-ethanol and EG-water mixtures with surfactants in the presence of sodium acetate as a reaction medium, yielding nanoparticles with a similar morphology to ours [47,48,49]. Magnetic properties vary, as it is possible to produce both superparamagnetic and soft ferromagnetic nanoparticles with a saturation magnetization ranging from 43 to 81 emu/g, residual magnetization from 0 to 8 emu/g, and coercivity varying within the range of 0–51 Oe, depending on the reaction conditions (Table 2). The zinc ferrite nanoparticles we have synthesized have superparamagnetic properties. Furthermore, based on the literature data, the spherical aggregates are much larger, with average diameters ranging from 110 to 345 nm, whereas the average sizes of the particles we synthesized are only 47 ± 17 and 63 ± 20 nm. The crystallite sizes determined by XRD are between 10 and 42 nm, while the sizes measured in our case are 9 ± 2 and 12 ± 2 nm, which fit well into the lower end of the range. Several additional synthesis methods suitable for the production of zinc ferrite nanoparticles are collected in Table 2; however, the corresponding magnetic properties (Ms, Mr, Hc) and particle morphology and size vary in a wide range.

The PVP stabilized composite contained 50 mg/g (5%) ferrite, and its solubility was 100 mg/5 mL (20 g/L) in distilled water, which gives a zinc ferrite concentration of 1 mg/mL, i.e., the stock solution used for the in vivo experiments.

In the field of MRI, the colloidal stability and average particle size profoundly influence nanoparticle properties across various imaging sequences [73]. Any occurrence of sedimentation, aggregation, or distortions in the stability and size escalation within a nanoparticulate system devised for MRI contrast proves detrimental. Such occurrences lead to inhomogeneities in the magnetic field, consequently impairing the imaging process. Thus, a meticulous control over colloidal stability and particle size is imperative to ensure the efficacy and precision of MRI techniques.

The hydrodynamic radius of the ferrite nanoparticles was measured using dynamic light scattering (DLS). The results (Figure 6) revealed distinct variations between the two samples composed of identical materials but prepared using different methods. The ZnFe_2_O_4_-NH_2_ Refl. sample exhibited a predominant presence of smaller sizes (hydrodynamic radius), while the ZnFe_2_O_4_-NH_2_ MW sample displayed larger sizes compared to the ZnFe_2_O_4_-NH_2_ Refl. sample. These results suggest that the preparation methods employed significantly influenced the size characteristics of the samples.

### 2.2. Results of the In Vitro MRI Measurements

The measurement of T2*, T2, and T1 relaxation times is the cornerstone in understanding tissue properties. T1 relaxation time represents the longitudinal or spin-lattice relaxation time, signifying the duration for the magnetization to recover approximately 63% of its longitudinal component after being disturbed by radiofrequency pulses. Unlike T2 and T2* relaxation times, which pertain to transverse relaxation, T1 relaxation time reflects the recovery of the magnetization along the longitudinal axis. This recovery is influenced by various tissue properties, including proton density and the mobility of water molecules within tissues [74]. In quantitative MRI studies, the interplay between T1, T2*, and T2 relaxation times is crucial for comprehensive tissue characterization. T1 contrast provides valuable information about the composition and integrity of tissues. For example, in clinical imaging, T1-weighted images are often used to highlight anatomical structures, as different tissues exhibit varying T1 relaxation times. Pathological conditions, such as tumors or lesions, can lead to alterations in T1 relaxation times, making T1 contrast an essential parameter for diagnostic imaging [74]. Comparatively, while T2* and T2 relaxation times focus on transverse relaxation and are influenced by magnetic field inhomogeneities, T1 relaxation time highlights the recovery of longitudinal magnetization and provides insights into the composition and structural integrity of tissues. Integrating information from T1, T2*, and T2 relaxation times offers a comprehensive view of tissue properties, enabling researchers and clinicians to gain valuable insights into various biological and pathological processes, making quantitative MRI a powerful tool in both research and clinical practice [75,76]. During the in vitro MRI measurements, five different ferrite concentrations (0.02, 0.05, 0.1, 0.2, and 1 mg/mL) in 2 mL Eppendorf tubes were used to measure the capabilities of the two samples (ZnFe_2_O_4_-NH_2_ Refl. And ZnFe_2_O_4_-NH_2_ MW) as MRI contrast agents. First, three fast spin echo scans were acquired with different sequence parameters to have T2-weighted, T1-weighted, and proton-density-weighted images.

Similar to the DLS measurements, there were variations detected between the ZnFe_2_O_4_-NH_2_ Refl. and ZnFe_2_O_4_-NH_2_ MW samples. In the case of MW samples, sedimentation was observed after the samples were placed in a strong magnetic field (Figure 7B), resulting in locally varying iron content and thus magnetic field inhomogeneity, which was impossible to shim properly. Distortions are observed in every FSE scan, as shown in Figure 7A. Additionally, there is a minor decrease in signal as the iron content increases. However, the accurate determination of iron concentration in the imaging plane is prevented by sedimentation, making quantitative relaxivity calculation unfeasible.

On the other hand, ZnFe_2_O_4_-NH_2_ Refl. samples could produce a homogenous MRI signal without any sign of sedimentation. Even at the lowest ferrite concentration, the transverse relaxation time (T2) decreased, causing a reduction in the MR signal observed on the T2-weighted FSE scan (Figure 8A). This phenomenon was also evident in the T1-weighted and PD-weighted scans, wherein a greater decrease in signal intensity was observed with increasing iron content than what would be expected from just the relaxation changes.

One of the most important characteristics of MRI contrast agents is relaxivity, their ability to change the relaxation time of the medium per unit concentration. The longitudinal relaxation times of each sample were determined based on the Multi-IR FSE scan, and the transverse relaxation times were derived from Multi-echo SE and Multi-echo GRE scans. The degree to which the relaxation rate (R1, R2, R2*), the inverse of the relaxation time, depends on the ferrite concentration gives the relaxivity of our sample. Figure 8B shows this linear relationship, and Table 3 contains the fitted relaxivity values. The measured values help to observe the r2/r1 ratio and determine the “true nature” of the contrast agent: the smaller the r2/t1, the more T1 effect is present in the sample.

The ZnFe_2_O_4_-NH_2_ Refl. sample (manufactured with the conventional, lengthy synthesis) has similar characteristics to SPIO nanoparticles; its transverse relaxivities are in the same range as Feraheme^®^ and Endorem^®^, but its r2/r1 ratio is four times higher, suggesting that this ferrite solution is almost only a T2 contrast agent and has an undetectable T1 effect.

### 2.3. In Vivo MRI Measurement

The ZnFe_2_O_4_-NH_2_ Refl. sample was chosen to be injected in vivo, as in the in vitro measurements, it was more stable than the MW sample. A dilution of 0.2 mg/mL ferrite concentration was available for injection, from which a dose of 1.3 mg/body weight kg was intravenously injected into the tail vein as a 0.2 mL bolus. Although the iron amount is 10–15 times smaller than usual in mouse SPIO measurements, the hypointense tissues are clearly defined. Immediate uptake is shown in the liver, and the ZnFe_2_O_4_-NH_2_ Refl. sample stays there long for at least a day (Figure 9). No other organ accumulates the sample according to our T2* map scans. The voxel-wise change of T2* values was determined for the first 8 min (right on Figure 9), in which bowel motion and liver uptake are clearly visualized.

According to the DLS measurement, the hydrodynamic size of ZnFe_2_O_4_-NH_2_ Refl. particles is in the range of SPIO nanoparticles, and the biodistribution of both are also the same—only the liver uptakes it, and the clearance is relatively slow.

For application in high-quality contrast MRI, the relatively long blood circulation time is essential, provided by the appropriate particles’ size. This was demonstrated using 4.0 nm PEGylated magnetic nanoparticles, which were suitable in a more than 2 h long contrast-enhanced angiography, while the 8.0 nm NPs with the same surface modification could only provide 30 min [73]. Despite the much larger (47 ± 14 nm) average size of our zinc ferrite nanoparticles, they still have a long retention time in the liver.

## 3. Materials and Methods

### 3.1. Materials

The zinc ferrite was synthetized from the following precursors: zinc(II) nitrate hexahydrate, Zn(NO_3_)_2_∙6H_2_O, molecular weight: 297.47 g/mol (Thermo Fisher GmbH, Kandel, Germany) nickel(II) nitrate hexahydrate, Ni(NO_3_)_2_∙6H_2_O, molecular weight: 290.79 g/mol (Thermo Fisher GmbH, Kandel, Germany), iron(III) nitrate nonahydrate, Fe(NO_3_)_3_∙9H_2_O (VWR International, Leuven, Belgium) and ethylene glycol, HOCH_2_CH_2_OH, (VWR Int. Ltd., Fontenay-sous-Bois, France), monoethanolamine, NH_2_C_2_H_4_OH (Merck KGaA, Darmstadt, Germany) and sodium acetate, CH_3_COONa (ThermoFisher GmbH, Kandel, Germany). For encapsulation of the Zn ferrite nanoparticles, poly(vinylpyrrolidone) K30 (PVP K30) (average molecular weight M.W. 50,000 g/mol) was used from Acros Organics Ltd. (Thermo Fisher Scientific, Geel, Belgium).

### 3.2. Characterization Techniques

High-resolution transmission electron microscopy (HRTEM, Talos F200X G2 electron microscope with field emission electron gun, X-FEG, accelerating voltage: 20–200 kV) was used for characterization of the particle size and morphology of the ferrite and palladium nanoparticles. For the imaging and electron diffraction, a SmartCam digital search camera (Ceta 16 Mpixel, 4k × 4k CMOS camera) and a high-angle annular dark-field (HAADF) detector were used. During sample preparation, the aqueous dispersion of ferrite was placed onto 300 mesh copper grids (Ted Pella Inc., Redding, CA, USA).

The phase analysis of the spinels was carried out with X-ray diffraction measurements using a Bruker D8 diffractometer (Cu-Kα source) in parallel beam geometry (Göbel mirror) with a Vantec detector. The average crystallite size of the domains was calculated by the mean column length calibrated method using the full width at half maximum (FWHM) and the width of the Lorentzian component of the fitted profiles. The identification of the surface functional groups of the ferrite nanoparticles was carried out with Fourier transformed infrared spectroscopy (FTIR) by applying the Bruker Vertex 70 equipment. During sample preparation, 10 mg ferrite was pelletized with 250 mg spectroscopic grade KBr; spectra were recorded in transmission mode.

The magnetic characterization of ferrite nanoparticles was carried out with a self-developed (University of Debrecen) vibrating-sample magnetometer system based on a water-cooled Weiss-type electromagnet. The powder samples were pelletized for the measurements with a typical mass of 20 mg. The magnetization (M) was measured as a function of the magnetic field (H) up to a 10,000 Oe field strength at room temperature. The magnetization of paramagnetic particles can be described by the Langevin function (Equation (3)).
(3)$$M(H)=M_{s}(\coth\left(\frac{m\mu_{0}H}{kT}\right)-\frac{1}{\frac{m\mu_{0}H}{kT}})$$
where M is the magnetization of the material, Ms saturation magnetization, Am^2^/kg (emu/g) of the material, m magnetic moment, Am^2^ (1 μB = 9.27401 × 10^−24^ Am^2^) of an individual particle.

DLS serves as a prevalent technique for quantifying the size distribution and colloidal stability of nanoparticles within a liquid suspension. By monitoring intensity fluctuations resulting from Brownian motion within the suspension, DLS elucidates crucial parameters: the hydrodynamic size (or apparent size), size distribution, and zeta potential. This method stands as a rapid means for quality assessment, offering insights into the colloidal properties of a liquid solution [77]. The hydrodynamic diameter of the particles was determined using a Litesizer 500 (Anton Paar, Hamburg, Germany). DLS measurements were performed at 25 °C in automatic mode (for backscatter detector, fixed at 175°; for side scatter, 90° detector angle; for front scatter, 15° detector angle) using a 633 nm He-Ne laser. Samples were measured in polystyrene disposable cuvettes (Anton Paar, Hamburg, Germany). During the measurements, the focus was kept steady. The measurement data were evaluated using software provided by the manufacturer, and statistical data and graphs were created and evaluated with Origin 9.0 (OriginLab) and Microsoft Excel 2013 software. For DLS (dynamic light scattering measurements) measurements, 100 mg of samples of ferrite content in the PVP support was 50 mg/g were used in vacuum vessels. For this, 5 mL of ultrafiltered water was added to obtain a solution with a concentration of 1 mg/mL. This solution was diluted 20-fold for the DLS measurements with ultrafiltered water.

MRI measurements were performed in vitro with a nanoScan^®^ PET/MR system (Mediso, Budapest, Hungary), having a 3 T magnetic field, 600 mT/m gradient system, and a volume transmit/receive coil with a diameter of 72 mm. In vitro scans were performed on five different ferrite concentrations (0.02, 0.05, 0.1, 0.2, and 1 mg/mL) of both samples (ZnFe_2_O_4_-NH_2_ Refl. and ZnFe_2_O_4_-NH_2_ MW) in 2 mL Eppendorf tubes. All FSE scans and relaxometry measurement were performed with the same geometrical parameters. One coronal slice was imaged with 4 mm of slice thickness, 50 mm of field of view, and 0.36 mm in-plane resolution. For determining T1 relaxation times, a Multi-IR FSE 2D sequence was used, with a repetition time of 5137 ms, echo time of 5.3 ms, and inversion times of 200, 400, 500, 600, 700, 800, 900, 1000, 1100, 1200, 1300, and 1600 ms. The total measurement time was 29 min.

The T2 relaxation times were determined using a Multi-echo SE 2D sequence with a repetition time of 3856 ms and a first echo time of 5.7 ms, followed by 31 echoes with echo intervals of 5.7 ms. The measurement time was 10 min. A Multi-echo GRE 2D sequence was used for the calculation of T2* relaxation times with a repetition time of 350 ms and a shortest echo time of 1.44 ms, which was followed by 11 echoes with echo intervals of 1.7 ms; the measuring time was 2 min. In vivo measurements were performed with the mice under isoflurane anesthesia (5% for induction and 1.5–2% to maintain the appropriate level of anesthesia; Arrane^®^, Baxter, Newbury, UK). The Multi-echo gradient echo (GRE) scans were collected at four different time points (pre-injection, 8 min, 3-, and 21-days p.i.). A 60 × 50 mm FOV on 46 coronal slices was acquired, with a matrix size 180 × 180, slice thickness of 0.5 mm, 4 averages, TR/first TE/FA 21 ms/1.82 ms/20°, and 8 echos with an echo space of 2.23 ms. The scan time was 15 min.

The relaxation time maps (T1, T2, and T2*) and region of interest (ROI) based evaluation were performed in Fusion (Mediso Ltd., Budapest, Hungary), and relaxivity calculations were done in Excel.

All procedures were conducted in accordance with the ARRIVE guidelines and the guidelines set by the European Communities Council Directive (86/609 EEC) and approved by the Animal Care and Use Committee of Semmelweis University.

### 3.3. Synthesis of the Amine-Functionalized Zinc Ferrite Nanoparticles

Amine-functionalized ZnFe_2_O_4_ spinels were synthesized using a modified coprecipitation method, employing two different synthesis methods (Figure 1). The ZnFe_2_O_4_-NH_2_ Refl. sample was prepared by refluxing and heating for 12 h (referred to as synthesis method “A”). In the first step, iron (III) nitrate nonahydrate (20 mmol) and zinc (II) nitrate hexahydrate (10 mmol), along with sodium acetate (150 mmol), were dissolved in 150 mL (2.7 mol) of ethylene glycol (EG). The solution was then heated to 100 °C in a round-bottom flask under reflux with continuous stirring under reflux. After 30 min, 35 mL of monoethanolamine (0.58 mol) was added. The solution was continuously agitated and refluxed for 12 h. After cooling, the solution was separated by centrifugation (4200 rpm for 10 min). The solid phase was washed several times with distilled water, and the magnetic ferrite was easily separated from the aqueous phase using a magnet. Finally, the ferrite sample was rinsed with absolute ethanol and dried overnight at 80 °C.

The coprecipitation method described above, which involves refluxing in ethylene glycol (EG) for 12 h, is a lengthy and energy-consuming synthesis method. Therefore, microwave synthesis was utilized to prepare amine-functionalized zinc ferrite, resulting in a notable reduction in the synthesis time to only 4 min (referred to as synthesis method “B”). During the experiment, the same quantities of metal precursors (20 mmol Fe(NO_3_)_3_ and 10 mmol Zn(NO_3_)_2_), sodium acetate (150 mmol), EG (150 mL), and MEA (35 mL) were used. A total of 30 mL-30 mL of the reaction mixture was measured in PTFE digestion tubes (with a volume of 50 mL). The reaction mixture was then placed in a CEM MDS 81 D microwave digestion instrument, where it was treated at 200 °C (350 W) for 4 min under atmospheric pressure. After cooling, the ZnFe_2_O_4_-NH_2_ MW sample was separated from the dispersion media and washed with distilled water.

To provide stability to the magnetic nanoparticles, PVP K30 was used to encapsulate them. During ultrasonication, 2.00 g of polyvinylpyrrolidone (PVP K30) was dissolved in 50 mL of distilled water. Then, 2 mL of the ZnFe_2_O_4_-NH_2_ dispersion (50 mg/mL) was added to the PVP solution. The mixture was sonicated for 4 min using a Hielscher UIP1000 Hdt tip homogenizer (130 W, 19 kHz). Afterwards, the colloid was dried using a rotary vacuum evaporator at 70 °C and 110 mbar. The resulting solid sample was then dried overnight at 80 °C. The as-prepared PVP-coated magnetic nanopowder had a ZnFe_2_O_4_ content of 50 mg/g. From this, 100 mg was measured and placed in vials. To obtain a colloid containing 1 mg/mL of ferrite, 5 mL of water was injected into the vials. This colloid was then ready to be tested as an MRI contrast agent.

## 4. Conclusions

Amine-functionalized zinc ferrite superparamagnetic nanoparticles were synthesized using a solvothermal method with two different synthesis techniques. The conventional solvothermal method, which involves a reflux step of 12 h, is a time-consuming and energy-intensive process. This synthesis was modified by applying a microwave treatment, a much faster and more economical route since the reaction time could be reduced to only 4 min for the formation of zinc ferrite nanoparticles. The morphological features of the two ferrite samples were the same, but there was a difference in particle size. The average particle diameters were 47 ± 14 nm (ZnFe_2_O_4_-NH_2_ Refl.) and 63 ± 20 nm (ZnFe_2_O_4_-NH_2_ MW), with the microwave synthesis resulting in larger nanoparticles. Phase identification measurements (XRD and SAED) confirmed that both magnetic samples contained only zinc ferrite, regardless of the different reaction conditions and techniques used. Moreover, the magnetic properties (based on VSM measurements) were virtually the same too. There was a significant difference in the zeta potential values of the two samples. The ZnFe_2_O_4_-NH_2_ Refl. samples had a zeta potential of −23.8 ± 4.4 mV, which was more negative compared to the +7.6 ± 6.8 mV measured for the ZnFe_2_O_4_-NH_2_ MW sample. The longer solvothermal synthesis led to the formation of more negatively charged ferrite particles. However, the colloidal systems’ stability was not long-lasting. To overcome the problem of colloidal stability, these ferrite nanoparticles were embedded in polyvinylpyrrolidone in a dried form. The magnetic nanoparticles we synthesized were embedded in PVP at a ferrite content of 50 mg/g. These polymer-stabilized red-brown crystalline samples could be easily redispersed in aqueous media. Two PVP-coated zinc ferrite samples were compared as contrast agents in magnetic resonance imaging (MRI). For the PVP-based ferrite samples, 100–100 mg amounts were dissolved in distilled water, resulting in the formation of stable ZnFe_2_O_4_ colloids at a concentration of 1 mg/mL.

An undiluted sample of ZnFe_2_O_4_-NH_2_ Refl. nanoparticles was administered via the tail vein of X BalbC mice. Nanoparticles lacking any conjugated specific in vivo targeting agent initially dispersed within the circulatory system and commenced accumulation primarily in the reticuloendothelial system (RES), such as the liver and spleen. The determined r1/r2 ratios were consistent with other SPION nanoparticles. Our sample exhibits a concentrated presence in the hepatic region of the animals, with comparable biodistribution and pharmacokinetics suspected.

Throughout our in vivo experiments, no toxic side effects were observed, and the mice were retained for further investigation in future studies.

The ZnFe_2_O_4_-NH_2_ Refl. sample’s nanoparticles, owing to their small (47 ± 14 nm) particle size, indicate a potential suitability for applications in the pharmaceutical industry. Smaller particle sizes can enhance dissolution rates, improve bioavailability, and enable more precise dosing, making the ZnFe_2_O_4_-NH_2_ Refl. Sample advantageous for pharmaceutical formulations. Although the preparation method for the ZnFe_2_O_4_-NH_2_ Refl. sample may be associated with lengthier and costlier processes, its significance in meeting quality criteria and providing informative results in the pharmaceutical industry justifies these considerations.

Conversely, the ZnFe_2_O_4_-NH_2_ MW sample’s size distribution, characterized by larger particles, suggests its suitability for industrial applications. The preparation method employed for the ZnFe_2_O_4_-NH_2_ MW sample proved efficient in generating particles of increased size. This attribute positions the ZnFe_2_O_4_-NH_2_ MW sample as a potential candidate for industrial use, particularly as a catalyst. Despite its larger size, the ZnFe_2_O_4_-NH_2_ MW sample retains acceptable properties for its intended use, indicating its viability in industrial settings where speed and efficiency are paramount.

## Figures and Tables

**Figure 1 ijms-24-16203-f001:**
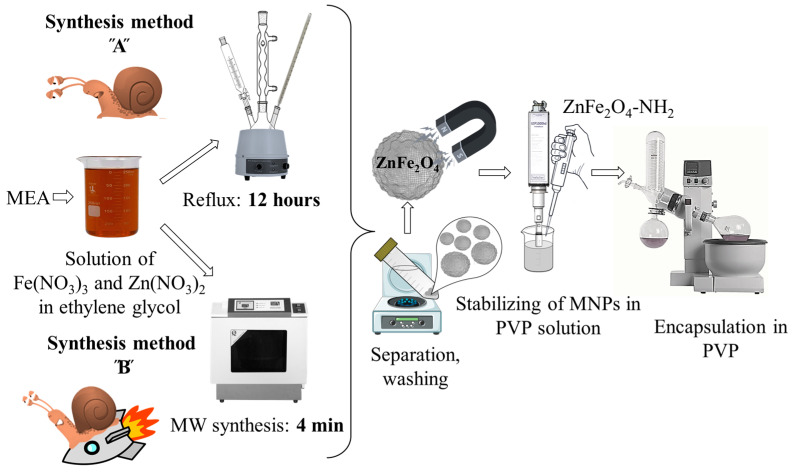
Essence of the preparation of NH_2_-functionalized zinc ferrite nanoparticles as MRI contrast agents. MEA and PVP stand for monoethanolamine and polyvinylpirrolidone, respectively.

**Figure 2 ijms-24-16203-f002:**
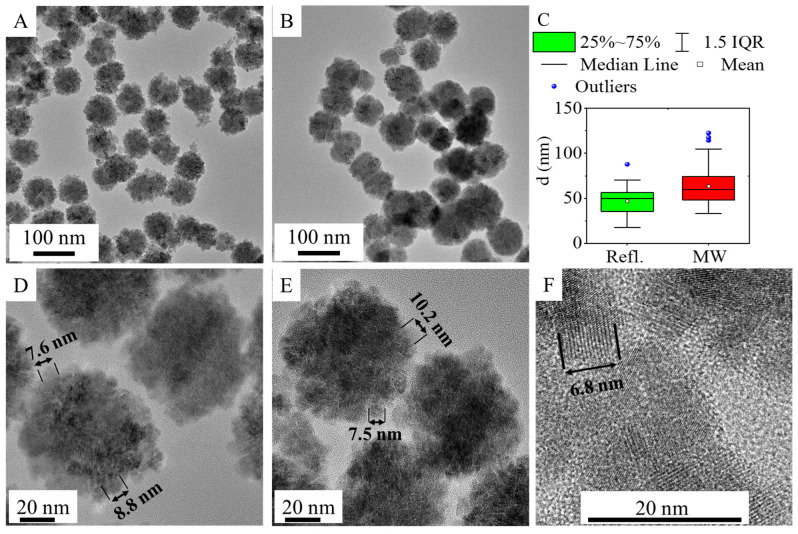
High-resolution transmission electron microscopy (HRTEM) pictures of ZnFe_2_O_4_-NH_2_ Refl. (**A**) and ZnFe_2_O_4_ MW (**B**) and their size distribution (**C**). High-resolution pictures with the individual spinel crystallites of ZnFe_2_O_4_-Refl. (**D**) and ZnFe_2_O_4_ MW (**E**,**F**).

**Figure 3 ijms-24-16203-f003:**
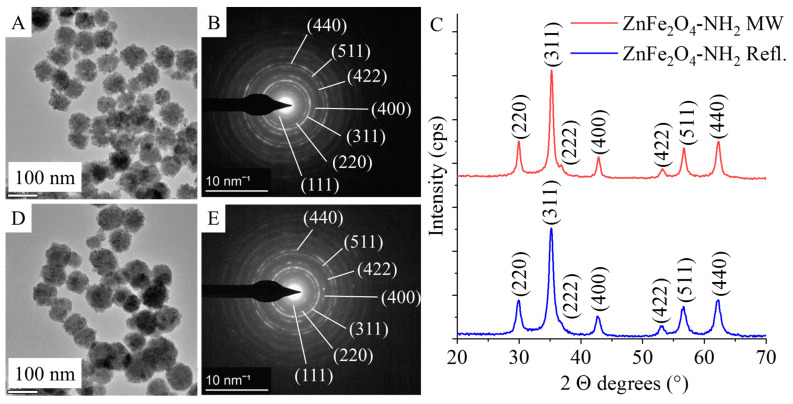
Transmission electron microscopy (TEM) and selected area electron diffraction (SAED) pictures of ZnFe_2_O_4_-NH_2_ Refl. (**A**,**B**) and ZnFe_2_O_4_-NH_2_ MW (**D**,**E**) and their XRD patterns (**C**).

**Figure 4 ijms-24-16203-f004:**
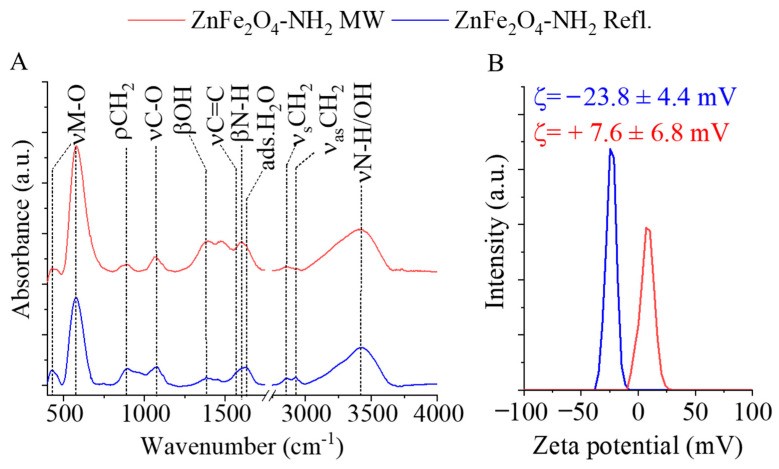
FTIR spectra (**A**) and zeta potential distribution (**B**) of the two amine-functionalized ZnFe_2_O_4_ samples.

**Figure 5 ijms-24-16203-f005:**
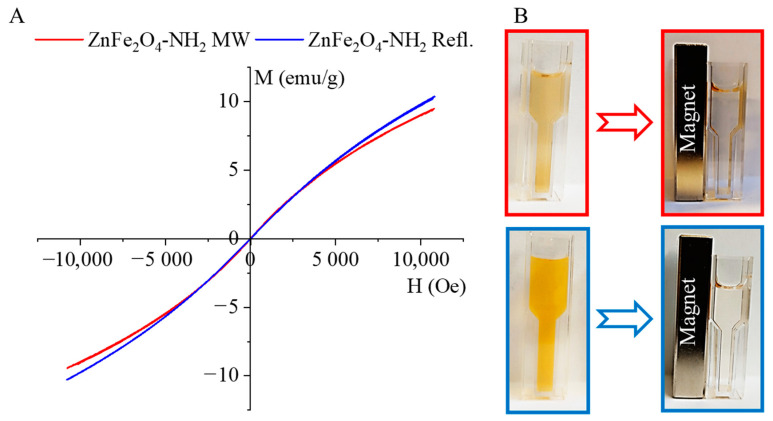
Magnetization curves (**A**) and magnetic separability (**B**) of the ZnFe_2_O_4_-NH_2_ samples.

**Figure 6 ijms-24-16203-f006:**
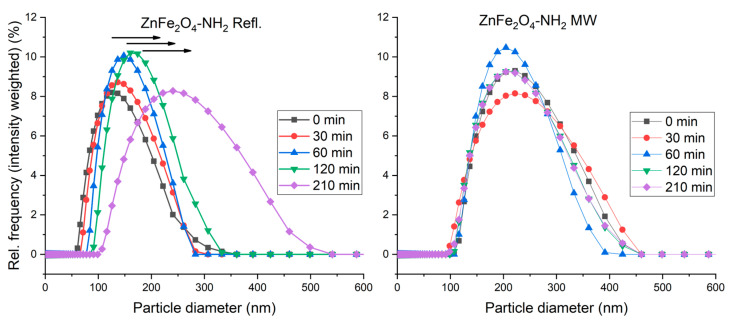
The size-distribution shift over time in the ZnFe_2_O_4_-NH_2_ Refl. and ZnFe_2_O_4_-NH_2_ MW samples. Based on these results, ZnFe_2_O_4_-NH_2_ MW has “better” colloidal stability over time.

**Figure 7 ijms-24-16203-f007:**
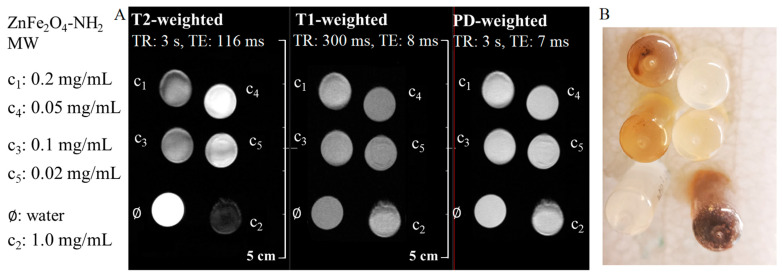
Summary of the produced MRI contrast enhancement in the aqueous dispersion of the ZnFe_2_O_4_-NH_2_ MW sample. (**A**) The fast spin echo sequences of T2-weighted, T1-weighted, and PD-weighted (proton density) images of the ferrite sample in different dilutions and dist. water (as pure solvent). (**B**) Sedimentation of MW samples after being placed in a strong magnetic field.

**Figure 8 ijms-24-16203-f008:**
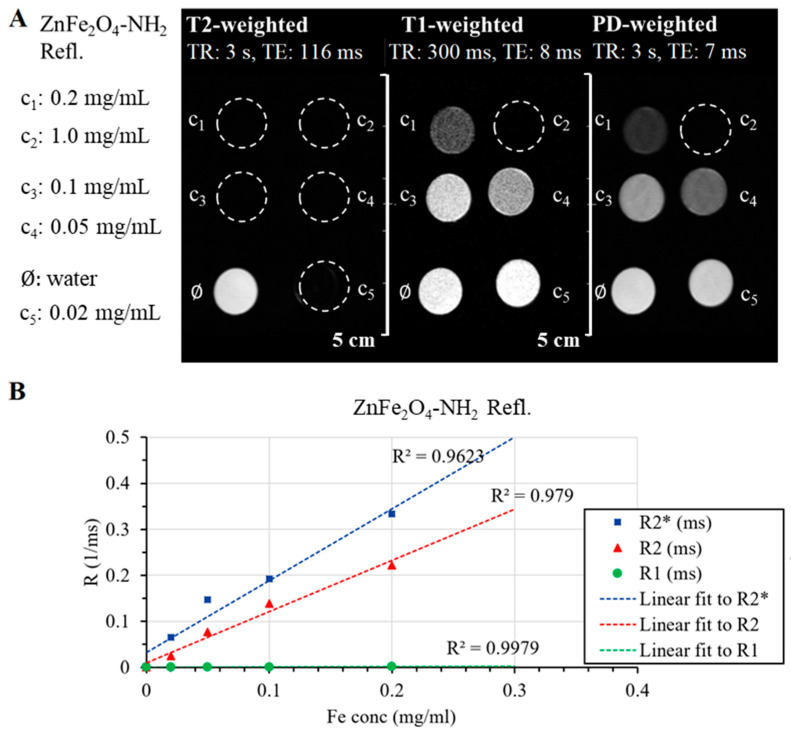
Summary of the produced MRI contrast enhancement in the aqueous dispersion of the ZnFe_2_O_4_-NH_2_ Refl. sample. The fast spin echo sequences of T2-weighted, T1-weighted, and PD-weighted (proton density) images of the ferrite sample in different dilutions and dist. water (as pure solvent) (**A**). Plot of transversal and longitudinal relaxation time changes with iron concentration (**B**). Linear fit was used for relaxivity calculations, and coefficients of determination (R^2^) are shown on the graph too.

**Figure 9 ijms-24-16203-f009:**
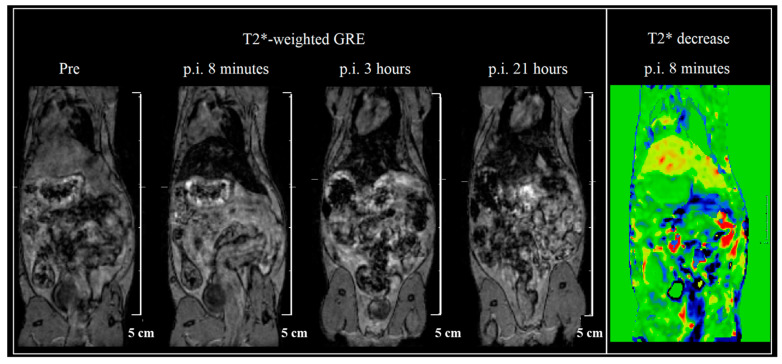
T2*-weighted gradient echo (GRE) scans of a mouse at 4 different timepoints—before injection and 8 min, 3 h, and 21 h after intravenous injection of ZnFe_2_O_4_-NH_2_ Refl. The voxel-wise change of T2* maps (5th image) shows the highest uptake in the liver. Liver signal intensity normed to a reference signal is shown in time on the last graph. T2* contrast refers to the decay of transverse magnetization seen with gradient-echo (GRE) sequences.

**Table 1 ijms-24-16203-t001:** Results of the size analysis (in nm) of the zinc ferrite nanospheres (based on HRTEM pictures).

(nm)	Mean	SD	Min.	Max.	P90	P95
ZnFe_2_O_4_-NH_2_ Refl.	47	14	18	88	64	66
ZnFe_2_O_4_-NH_2_ MW	63	20	33	123	88	105

**Table 2 ijms-24-16203-t002:** Comparison table for magnetic and morphological properties of ZnFe_2_O_4_ nanoparticles synthesized by different methods.

Synthesis Method	Ms (emu/g)	Mr (emu/g)	Hc (Oe)	Crystallite Size XRD (nm)	Particle Size EM (nm)	Reference
Solvothermal	18.35	0	0	9 ± 2	47 ± 17	This work
	16.99	0	0	12 ± 2	63 ± 20	
Solvothermal	60.4	0.83	9.9	25.3	150 ± 25	[47]
	60.3	0.18	0.2	15.8	130 ± 30	
	52	1.31	22.0	20.4	120 ± 30	
	43.2	0.35	2.1	15.1	300 ± 50	
Solvothermal	66.71	0.42	3.67	12.2	345.2	[48]
	58.46	0.30	3.15	11.1	340.8	
	66.52	0.38	3.52	10.0	312.8	
Solvothermal	66.71	0	0	12.9	345.2	[49]
	81.34	5.2	34.60	40.7	150.6	
	76.65	7.5	51.24	24.5	110.6	
Solvothermal (reflux)	50.4	0	0	11	10	[65]
Ball milling	30	0	0	16	n.d.	[66]
Hydrothermal	10	2	100	26	n.d.	[67]
Sol–gel	12.9	3.4	352.1	22	n.d.	[68]
Microwave combustion	2.6	0.01	7.5	37.6	n.d.	[69]
Microemulsion	1.49	negl.	negl.	15.3	n.d.	[70]
Thermal decomposition	43	negl.	negl.	9.8	9.8	[71]
Coprecipitation	2.51	0.44	69.83	34	30–50	[72]
	2.31	0.22	60.59	36.3		

**Table 3 ijms-24-16203-t003:** Longitudinal (r1) and transversal (r2 and r2*) relaxivity of ZnFe_2_O_4_-NH_2_ Refl.

(mL/mg/ms)	r1	r2	r2*
ZnFe_2_O_4_-NH_2_ Refl.	0.007	1.113	1.562

## Data Availability

Data are available upon request from the corresponding authors.

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
