# Peer review of "Development of Polymer-Encapsulated, Amine-Functionalized Zinc Ferrite Nanoparticles as MRI Contrast Agents"

_ijms, 2023, doi:10.3390/ijms242216203_

Round 1

Reviewer 1 Report

Comments and Suggestions for Authors

The manuscript entitled, ‘Development of Polymer Encapsulated, Amine-Functionalized Zinc-Ferrite Nanoparticles as MRI Contrast Agent’ reported magnetic nanoparticles for MRI applications. The article should be modified according to the following points;

1.      Figure 5 shows superparamagnetic behavior. This should be elaborate more.

2.      The introduction is not detailed. Some references should be incorporated and the novelty of the system should be reported.

3.      Why ZnFe2O4 spinels are more significant compared to the normal magnetic nanoparticles?

4.      Some articles have significance (related to sensing) and could be discussed with the help of following references:

(a)    Ganguly, S., Neelam, Grinberg, I., & Margel, S. (2021). Layer by layer controlled synthesis at room temperature of trimodal (MRI, fluorescence and CT) core/shell superparamagnetic IO/human serum albumin nanoparticles for diagnostic applications. Polymers for Advanced Technologies32(10), 3909-3921.

(b)   Ganguly, S., & Margel, S. (2021). Design of magnetic hydrogels for hyperthermia and drug delivery. Polymers13(23), 4259.

(c)    Chen, C., Ge, J., Gao, Y., Chen, L., Cui, J., Zeng, J., & Gao, M. (2022). Ultrasmall superparamagnetic iron oxide nanoparticles: A next generation contrast agent for magnetic resonance imaging. Wiley Interdisciplinary Reviews: Nanomedicine and Nanobiotechnology14(1), e1740.

Author Response

Thank you very much for taking the time to review this manuscript. Please see the attachment.

Reviewer 2 Report

Comments and Suggestions for Authors

In the present work, authors reported the synthesis of amine-functionalized zinc ferrite superparamagnetic nanoparticles by using two different solvothermal methods, and then investigated their structural, morphological, and magnetic resonance properties. Results indicated that the sample exhibited a concentrated presence in the hepatic region of the animals, with comparable biodistribution and pharmacokinetics suspected. Overall, the performance sounds good, and results are clearly presented. However, some issues should be addressed.

1, Authors may rearrange/polish the text and elaborated " Materials and Methods  " section the way so anybody can repeat the procedures, like a recipe.

2, The formation mechanisms of two different ferrite particles were not well introduced. Authors should give more detail description about the formation mechanisms. Some key and important research results in ferrite field should be mentioned and cited, such as Journal of Materials Chemistry C, 2016, 4, 9738; Nano-Micro Letters, 2011, 3, 91.

3, Lots of articles have been already published on ZnFe2O4 nanoparticles. In such case, author should compare and discuss the previous results on ZnFe2O4 nanoparticles. Make a results Table of various parameters.

4, The particle size was hardly observed from the HR-TEM image. It is better to provide high-resolution image to replace the old one.

5, The mechanisms in the MRI measurements were not well explained. The discussions correlating the mechanisms and DLS measurement should be deepened for clarity.

Author Response

(The authors gave the same response as above.)

Round 2

Reviewer 1 Report

Comments and Suggestions for Authors

This can be published in its present form.